# Oxidative Stress in the Pathogenesis of Oral Cancer

**DOI:** 10.3390/biomedicines12061150

**Published:** 2024-05-23

**Authors:** Cătălina Ionescu, Fatima Zahra Kamal, Alin Ciobica, Gabriela Halitchi, Vasile Burlui, Antoneta Dacia Petroaie

**Affiliations:** 1Department of Biology, Faculty of Biology, Alexandru Ioan Cuza University of Iasi, Bd. Carol I no. 20A, 700505 Iasi, Romania; catalinaionescu81@yahoo.com (C.I.); alin.ciobica@uaic.ro (A.C.); 2Clinical Department, Apollonia University, Păcurari Street 11, 700511 Iasi, Romania; vburlui@gmail.com; 3Higher Institute of Nursing Professions and Health Technical (ISPITS), Marrakech 40000, Morocco; 4Laboratory of Physical Chemistry of Processes and Materials, Faculty of Sciences and Techniques, Hassan First University, B.P. 539, Settat 26000, Morocco; 5Center of Biomedical Research, Romanian Academy, Iasi Branch, Teodor Codrescu 2, 700481 Iasi, Romania; 6Academy of Romanian Scientists, Str. Splaiul Independentei no. 54, Sector 5, 050094 Bucharest, Romania; 7Faculty of Medicine, Grigore T. Popa University of Medicine and Pharmacy, 700115 Iasi, Romania; pantoneta@yahoo.com

**Keywords:** oxidative stress, oral cancer, oral squamous cell carcinoma, oral cancer patients, neuropsychiatric complications, oral cancer biomarkers

## Abstract

Oxidative stress, arising from an imbalance between reactive oxygen species (ROS) and antioxidants, contributes significantly to oral cancer such as oral squamous cell carcinoma (OSCC) initiation, promotion, and progression. ROS, generated both internally and externally, induce cellular damage including DNA mutations and lipid peroxidation, fostering oncogene activation and carcinogenesis. The objective of this review was to cover and analyze the interplay between ROS and antioxidants, influencing the key processes such as cell proliferation, apoptosis, and angiogenesis, shaping the trajectory of OSCC development. Despite the promise of antioxidants to halt cancer progression and mitigate oxidative damage, their therapeutic efficacy remains debated. The conducted literature search highlighted potential biomarkers that indicate levels of oxidative stress, showing promise for the early detection and monitoring of OSCC. Furthermore, melatonin has emerged as a promising adjunct therapy for OSCC, exerting antioxidant and oncostatic effects by modulating tumor-associated neutrophils and inhibiting cancer cell survival and migration. In addition, this review aims to shed light on developing personalized therapeutic strategies for patients with OSCC such as melatonin therapy, which will be discussed. Research is needed to elucidate the underlying mechanisms and clinical implications of oxidative stress modulation in the context of oral cancer.

## 1. Introduction

Physiological systems have evolved with a multitude of complex biochemical cycles that are regulated at several levels to optimize the production of valuable metabolites in the cell. The critical regulation of these cycles is essential to prevent the over-production of cellular components. It is known that over-exposure to any compound triggers side effects. The best example to explain this scenario in physiological systems is the role of free radicals. At optimum concentrations, they act as the backbone of the immune system and form the key component of the phagocytic cells and apoptotic processes. However, when their concentration is elevated, they cause oxidative stress, leading to the oxidation of proteins, lipids, and DNA. The resulting changes in the structure of macromolecules affect their functioning. In turn, these events proceed toward the misinterpretation of protein translation mechanisms and genetic information, giving rise to a number of human diseases including cancer [1,2].

Free radicals are defined as “a molecular species that can exist independently and contains unpaired electrons in their atomic orbital”. The unpaired electrons can either accept another electron or they can be donated. Thus, these molecules are extremely unstable, highly reactive, and can act as either oxidants or reductants [3]. In physiological systems, they occur in the form of hydroxyl radicals (OH^−^), superoxide anion radicals (O_2_^−^), hydrogen peroxide (H_2_O_2_^−^), singlet oxygen (^1^O_2_^−^), hypochlorite (CIO^−^), nitric oxide radicals (NO), and peroxynitrite radicals (ONOO^−^).

The free radicals are generated as a by-product of normal metabolic processes in humans. In addition, various pollutants can cause the excess production of reactive species. A few examples of these agents include ozone, cigarette smoke, radiation, pesticides, and industrial chemicals. On exposure to these pollutants, several enzymatic and non-enzymatic reactions are triggered, which increase the rate of phagocytosis, prostaglandin synthesis, and cytochrome P-450 respiration. The basis of all of these mechanisms rests on the induction of free radical synthesis in mitochondria and peroxisomes [4]. The adverse effects caused by free radicals progressively accumulate in the cells over a long period of time in the form of multiple small changes. Along with these changes, combined with genetic and environmental factors, they lead to chromosomal defects. The major disease is linked to the pathological effects of free radicals in cancer [5].

Tumor promotion and a pro-oxidant state association are well-established, with ROS playing a significant role in the progression of carcinogenesis. Cancer development is a complex process involving three distinct stages: initiation, promotion, and progression, each influenced by oxidative stress and ROS activity. During the initiation phase, normal cells undergo DNA mutations, leading to the emergence of initiated cells with permanent genetic alterations. ROS-induced DNA modifications are observed in cancer tissues, highlighting the involvement of oxidative stress in the initiation of carcinogenesis. In the promotion phase, initiated cells undergo rapid multiplication fueled by increased cell proliferation or the suppression of apoptosis. Oxidative stress contributes significantly to this phase by temporarily influencing genes associated with cell division and programmed cell death. Additionally, ROS modulate the function of key transcription factors such as nuclear factor-κB (NF-κB), Nrf2, hypoxia-inducible factor (HIF), and p53, which regulate cellular growth and cancer initiation. Even minimal oxidative stress during this phase can trigger cell division, promoting tumor progression primarily through ROS generation. Furthermore, ROS play a crucial role in the progression phase of carcinogenesis. Elevated ROS production contributes to diverse processes such as mutations, inhibition of antiproteases, upregulation of matrix-metalloproteinases (MMPs), and damage to neighboring tissues. Elevated levels of oxidatively modified DNA bases exacerbate genetic instability and enhance the metastatic potential of established cancer cells. ROS also play a pivotal role in pro-angiogenic response, which is crucial for cancer metastasis. Across all stages of carcinogenesis, ROS interact with cellular components, contributing to neoplastic transformation through diverse pathways and events [6].

## 2. Overview on Oxidative Stress, Cancer, and Antioxidants

The identification of free radicals as the causative agent of oxidative stress and triggering factors for the development of cancer has led to the exploration of agents that can neutralize these reactive species. In the early nineteenth century, the beneficial role of antioxidants was recognized. However, it was during the later period of the same century that the functional role of vitamins (A, C, and E) was demonstrated as antioxidants, which completely revolutionized the medical field. This was because it led to the possibility of literally scavenging the free radicals in the biological systems before they reached a concentration detrimental to cellular processes [2].

With the advances in the medicinal field, we now know that cancer pathology is a complex process arising due to intrinsic cellular and molecular damage. We also know that the initiation of cancer by the activation of oncogenes can be induced by free radicals due to their ability to cause mutation, transformation, and the activation of carcinogenic factors. The antioxidants prevent oxidative stress by neutralizing the ROS and/or inhibiting the proliferation of stressed cells. Basically, the antioxidants simply donate an electron to the free radical in order to stabilize them. Alternatively, they act as catalysts to catabolize the free radicals into non-reactive compounds through a series of enzymatic processes. Hence, they are known as enzymatic antioxidants. Examples include superoxide dismutase (SOD), catalase (CAT), and glutathione peroxidase (GPx). Working synergistically, these antioxidants counteract the detrimental effects of free radical-induced oxidative stress by scavenging ROS and detoxifying their by-products. Vitamins are non-enzymatic antioxidants. Vitamin C neutralizes free radicals by enhancing the immune response and inducing the production of detoxification enzymes in the liver. Vitamin E promotes both humoral and cell mediated immunity to repair DNA damage [7]. A few other examples of naturally occurring non enzymatic antioxidants are ubiquinol, melatonin, B-carotene, and uric acid.

ROS play a significant role in the multi-step process of carcinogenesis, highlighting their importance in various types of cancer including oral cancer. While oral cancer can be easily prevented by eliminating the majority of risk factors, it can still occur in individuals outside high-risk categories. Initially, the oral cavity was primarily considered as a source of pathogenic microorganisms and antigens. However, recent studies suggest that it may also contribute to systemic oxidative stress and inflammation [8].

In response to the damaging effects of free radicals on cellular components including proteins, membrane lipids, and other biomolecules, mammalian cells activate a robust antioxidant defense system. This defense system comprises enzymatic antioxidants such as superoxide dismutase (SOD), catalase (CAT), and glutathione peroxidase (GPx) as well as non-enzymatic antioxidants like reduced glutathione (GSH). Working synergistically, these antioxidants counteract the detrimental effects of free radical-induced oxidative stress by scavenging ROS and detoxifying their by-products. This includes lipid peroxides, protein carbonyls, and other oxidative modifications. The cascade of antioxidant enzymes, particularly SOD, CAT, and GPx, plays a crucial role in scavenging ROS, thus preventing oxidative damage to cellular components. By neutralizing these by-products, the antioxidant defense system helps maintain cellular integrity and reduces the risk of cellular transformation into a malignant phenotype [9].

The biochemical activity of antioxidant transformation products is equally important. When antioxidants endure metabolic transformations in the body, they generate a diversity of metabolites with distinct biological activities. These by-products result from reactions with radicals and hydroperoxides and exhibit different antioxidant activities. Antioxidants, which include organic compounds of sulfur and phosphorus, undergo transformation during the hydroperoxide decomposition process. This process leads to various oxidation products being formed like sulfoxides, S-centered radicals, and sulfenic acid. Phenols and aromatic amines, on the other hand, trap alkyl and alkoxyl radicals by chain-breaking mechanisms. These antioxidants produce transformation products such as phenoxy and aminyl radicals, which then react to form alkylperoxycyclohexadienones (R02-CHD) and other compounds [10,11]. The resulting transformation products show anticancer properties. Sulforaphane, a sulfoxide compound found in cruciferous vegetables, has shown potential in cancer prevention and treatment. It has been found to induce apoptosis, arrest cell growth, and modulate carcinogen metabolism [12,13]. Additionally, it has been shown to have anticancer effects including protection from DNA damage and modulation of the cell cycle [14]. These effects are mediated, at least in part, by the activation of the transcription factor Nrf2 [15]. However, the influence of sulfonylurea derivatives, another type of sulfoxide, on cancer risk is less clear, with conflicting results in studies [16]. In addition, a range of alkylperoxycyclohexadienones have been investigated for their potential in cancer therapy. Eibl et al. [17] and Sugiura et al. [18] both explored the antineoplastic proprieties of these compounds, with Eibl focusing on hexadecylphosphocholine and Sugiura on 5-alkylidene-4-hydroxy-2-cyclopentenones. Chen et al. [19] synthesized and evaluated a series of 2-alkylaminomethyl-5-(E)-alkylidene cyclopentanone hydrochlorides, finding them to be active against various human cancer cell lines. Veinberg et al. [20] further expanded the range of alkylperoxycyclohexadienones, synthesizing 6-alkylidenepenicillanate sulfones and related 3-alkylidene-2-azetidinones, which showed potent cytotoxic properties toward tumor cells.

Today, there is enough scientific evidence to point to the paradox of antioxidants. They have been considered a double-edged sword, capable not only of inhibiting but also promoting carcinogenesis, depending on their concentration, balance with pro-oxidants, and context of use. While studies have highlighted the ability of antioxidants to restore oxidative balance and protect against oxidative stress-induced DNA damage, other studies have investigated the ability of antioxidants to fuel carcinogenesis. Banerjee et al. noted that the presence of N-acetyl cysteine (NAC), an exogenous antioxidant derived from the amino acid cysteine and commonly utilized as precursor in synthesizing glutathione, diminished the proapoptotic impact of andrographolide on colon cancer cell lines (T84 and COLO 205). The proapoptotic effect of andrographolide hinges on inducing endoplasmic reticulum (ER) stress through the production of reactive oxygen species (ROS) [21]. In addition, an antioxidant excess has been found to compromise the elimination of cancer cells during radiation therapy by potentially inhibiting the mechanisms of cytotoxicity triggered by oxidative stress, which may influence the outcome of carcinogenesis [22]. A study by Sakamoto et al. indicated that vitamin E decreased the radiation effectiveness on both normal and malignant cells [23]. Similarly, Witenberg et al. found that vitamin C alleviated radiation-induced apoptosis in cancer cells [24]. These findings suggest that antioxidants may hinder the intended cytotoxic effects of radiation on cancer cells. Furthermore, a study by Sayin et al. found that adding antioxidants to the diet accelerated lung cancer progression by reducing the ROS levels, preventing p53 activation and promoting tumor cell proliferation. In B-RAF- and K-RAS-induced lung cancer mouse models, supplementation with two antioxidants, N-acetylcysteine (NAC) and vitamin E, significantly increased tumor burden and reduced survival rates. RNA sequencing analysis revealed that despite their structural differences, both NAC and vitamin E induced closely coordinated alterations in the tumor transcriptome profiles. These alterations were mainly characterized by an increase in the proliferation of cells expressing the wild-type form of the p53 protein, but not those with mutations in this protein [25]. Excessive intake of antioxidants can also disrupt the oral microbiome, impede wound healing, and cause cytotoxic effects, leading to oral tissue damage [26]. Antioxidant-rich foods and beverages, particularly those high in acidity, can contribute to dental enamel erosion, weakening teeth and increasing susceptibility to cavities and sensitivity [27]. In addition, antioxidants can potentially protect cancer cells from oxidative damage, reduce the effectiveness of cancer treatments, and inhibit apoptosis, thereby promoting the survival and growth of oral cancer cells [28]. However, the specific harmful effects of antioxidants on the oral area have not been thoroughly addressed in studies yet. It is important to balance antioxidant intake and consult healthcare professionals to avoid adverse effects on oral health [29].Topical antioxidant remedies can be beneficial in reducing free radicals or reactive oxygen species in the oral environment [30].

## 3. Oxidative Stress and Antioxidant Mechanism in Oral Cancer Research: Insights from Epidemiological, Clinical, and Experimental Studies

Oral cancer encompasses malignancies affecting the lip, various areas within the mouth, and the oropharynx, collectively ranking as the 13th most prevalent cancer globally, as established by the World Health Organization. The worldwide incidence of lip and oral cavity cancers was estimated at 377,713 new cases and 177,757 deaths in 2020. This type of cancer exhibits a higher occurrence in men and tends to be more lethal among men than women. Additionally, the incidence of oral cancer shows significant variation based on socio-economic circumstances, with higher rates observed in older individuals [31].

Oral squamous cell carcinoma (OSCC) stands as the most predominant malignant disease that develops in the oral cavity, making up the vast majority, around 90%, of all oral cancers. Despite ongoing progress in surgical techniques and other treatment modalities, the mortality rate associated with oral cancer has largely remained constant over the years [31,32].

Many studies have investigated the link between oxidative stress and oral cancer. This process is mediated through various mechanistic pathways that contribute to cancer initiation and progression. The overexpression of enzymes like NOX, COX, LOX, and NOS are significant sources of ROS/RNS in oral cancer [33,34,35,36,37] These species act as redox messengers, impacting the activity of kinases/phosphatases and signal transduction pathways, which in turn regulate transcription factors involved in cancer development. In addition, ROS/RNS modulate the activity of a range transcription factors, including APE1/Ref-1, HIF-1α, AP-1, Nrf2, NF-κB, p53, FOXO, STAT, and β-catenin [37]. These factors are essential in genes associated with cell proliferation regulation, survival, and apoptosis, thereby influencing cancer progression [37]. Furthermore, the canonical WNT/β-catenin pathway plays a significant role in oral cancer, influencing prognosis and treatment response [38,39]. Its aberrant activation can lead to apoptosis resistance and tumorigenesis [39], while its role in oral carcinogenesis is well-documented [40]. The wingless-type-1 (WNT-1) pathway, a key component of this signaling, has been implicated in the progression and metastasis of oral cancer [41] Activation of WNT signaling begins with the WNT ligand binding to Frizzled receptors and LRP5/6 co-receptors, triggering beta-catenin stabilization and its subsequent accumulation in the cytoplasm. Translocation of beta-catenin into the nucleus forms a complex with TCF/LEF transcription factors, activating target genes like C-Myc, cyclin D1, c-jun, fra-1, and u-PAR, implicated in tumor progression. Dysregulation of this pathway in oral cancer correlates with heightened aggressiveness, metastatic propensity, and poorer prognosis, underscoring its significance as a therapeutic target for OSCC management [41,42,43].

Lifestyle interventions significantly impact cancer treatment outcomes, both positively and negatively [44].

The primary risk factors for OSCC, accounting for over 90% of cases, are prolonged alcohol consumption and tobacco use. Tobacco smoke contains three chemical groups—nitrosamines, benzopyrenes, and aromatic amines—that promote cancer. Smokers face a threefold higher risk of developing OSCC compared to nonsmokers. Even for nonsmokers, exposure to secondhand smoke increases the expectation of developing OSCC by 87% compared to those unexposed. Moreover, smoking not only compromises the oral cavity’s immunity, but also worsens conditions like gingivitis, periodontitis, and, notably, OSCC. Known as ethanol, alcohol has harmful effects on the organism. At the local level, it increases permeability into the oral mucosa, dissolves lipid particles of the epithelium, and induces general epithelial atrophy. Systemically, it exerts a mutagenic effect, reducing salivary flow and compromising the liver’s ability to handle carcinogenic chemicals. This systemic impact eventually impairs the immune system, increasing the risk of infections and the development of abnormal tissue growth. Other risk factors may include ultraviolet radiation, particularly in the context of lip cancer. However, lip cancer, along with other factors, has been linked to additional contributors such as human papillomavirus (HPV) and Candida infections, nutritional deficiencies, and genetic predisposition [45,46].

Smoking is strongly associated with various cancers development, with nine out of ten lung cancer deaths. Studies indicate that smoking can alter the metabolism of anticancer medications, potentially leading to higher clearance rates of specific drugs like erlotinib, which could diminish treatment efficacy. In addition, smoking may also affect the metabolism of irinotecan, an antineoplastic agent that acts as a specific inhibitor of DNA topoisomerase, leading to lower concentrations of its active metabolite, SN-38, and potentially altering treatment outcomes [44]. Another treatment-disturbing factor is alcohol. Moderate alcohol consumption has been linked to reduced overall mortality, but heavy alcohol intake is associated with a higher risk of developing specific cancers including head and neck cancer and esophageal carcinoma. Similar to smoking, continued alcohol use after a cancer diagnosis or treatment is associated with a higher risk of disease recurrence or second malignancy. Alcohol consumption can influence the activity of drug-metabolizing enzymes, particularly CYP3A, a human gene locus responsible for the metabolism of more than 50% of all therapeutic medications that undergo phase I metabolism, which may impact the metabolism and efficacy of certain anticancer drugs [44]. Healthcare providers are key figures in educating patients about the dangers linked to tobacco and alcohol consumption concerning OSCC. Promoting smoking cessation and moderating alcohol intake can substantially decrease the risk of OSCC. Additionally, regular oral health screenings should be emphasized for early detection and intervention. Some studies emphasize the importance of this education, specifically highlighting the effectiveness of personalized reports in reducing alcohol-related risks [47,48]. Patton et al. [49] underscored the need for training in tobacco and alcohol cessation, particularly among dental providers. Shaik et al. [50] further underscored the importance of oral health professionals in advocating for tobacco-free lifestyles and implementing intervention programs.

On the other hand, lifestyle interventions that include exercise and dietary advice have shown feasibility and positive impacts on health outcomes such as exercise behavior, fatigue, and functional capacity in cancer survivors [51]. In their single arm feasibility study, Vassbakk-Brovold et al. aimed to evaluate the feasibility of a 12-month lifestyle intervention program (like diet, physical activity, stress management, and smoking cessation) for cancer patients undergoing chemotherapy. Participants were newly diagnosed cancer patients receiving chemotherapy with curative or palliative intent at a cancer treatment center in Norway. Out of 161 eligible patients, 100 agreed to participate, with a completion rate of 38%. Dropouts were associated with older age, certain diagnoses, and smoking status. A significant dietary habits, distress levels, and physical activity improvements was recorded by the finding as well as the feasibility and potential benefits of a comprehensive lifestyle intervention for cancer patients undergoing chemotherapy including those in palliative care [52]. In addition, Godsey et al. explored the potential synergy between herbal supplements and conventional treatments. For instance, anthocyanins extracted from black raspberries have exhibited notable cancer-preventing properties in both laboratory and animal studies. Application of an anthocyanin gel to the oral cavity has shown promise in suppressing genes associated with cancer cell proliferation as well as inhibiting esophageal tumorigenesis and providing protection against benzo(a)pyrene-induced transformation [53]. Green tea polyphenols have been also extensively studied for their chemo preventive potential. They trigger apoptosis, induce cell cycle arrest, and block the epidermal growth factor receptor tyrosine kinase phosphorylation. (−)-Epigallocatechin-3-gallate (EGCG), a green tea polyphenol, has shown promising results in inhibiting carcinogenesis in animal models [54]. These polyphenols have also been linked to a reduced risk of various cancers including oral cancer in epidemiological studies [55]. In a mouse model of prostate cancer, the oral consumption of green tea polyphenols was found to inhibit insulin-like growth factor-I-induced signaling, suggesting a potential mechanism for their anticancer effects [56]. These studies collectively suggest that lifestyle interventions including the avoidance of harmful habits and the use of complementary treatments can significantly affect the effectiveness of cancer treatment.

The understanding of the role of oxidative stress in oral carcinogenesis highlights the potential of antioxidant-based interventions in preventing, inhibiting, and even reversing OSCC progression. The potential for antioxidant-based interventions in preventing, inhibiting, and even reversing OSCC progression is highlighted, with a focus on the role of antioxidant phytochemicals in periodontal disease [57]. Regular monitoring of antioxidant levels and oxidative stress markers is recommended to assess disease progression and treatment response [6]. Healthcare providers may consider incorporating antioxidant supplementation or dietary recommendations into OSCC management strategies [58].

Numerous researchers have extensively documented the pivotal role of ROS in both initiating and promoting the multistep process of carcinogenesis. To counteract the detrimental effects of these free radicals, all aerobic cells are equipped with robust antioxidant defense mechanisms. Oral precancer and cancer are characterized by heightened levels of ROS and reactive nitrogen species (RNS), coupled with diminished antioxidant levels. Various antioxidants offer protection against the harmful effects of these free radicals. Treatment interventions utilizing antioxidants hold promise in potentially preventing, inhibiting, and even reversing the multiple stages of oral carcinogenesis [59].

The landscape of oxidative stress in oral cancer research is illuminated by a diverse array of studies spanning epidemiological investigations, clinical trials, and experimental endeavors. Several studies have highlighted the association between oral cancer and both oxidative stress and compromised antioxidant defenses. Korde et al. [60] recorded a significant increase in the malondialdehyde (MDA) (*p* = 0.036) and total nitric oxide (TNO) (*p* = 0.012) levels as well as a decrease in total antioxidant capacity (TAC) (*p* = 0.000) in both the serum and tissue of oral cavity cancer patients compared to controls. Furthermore, the positive correlation observed between the TNO and MDA levels (r = 0.42, *p* = 0.01), which intensified with OSCC grade, underscores the complex relationship between reactive nitrogen species (RNS)-mediated lipid peroxidation and antioxidant status in oral cancer progression (Table 1) [60].

Similarly, the serum levels in the oral carcinoma patients showed a significant decrease in iron, a cofactor of the enzymes catalase and peroxidase, and selenium, a cofactor of the enzymes glutathione and peroxidase. This drop in serum selenium and iron levels detected in patients with oral carcinoma could be due to increased oxidative stress or inadequate dietary intake. Enzymatic processes that depend on selenium and iron as cofactors could exacerbate this decline. This finding highlights the significant potential of antioxidant supplementation or dietary intake in the prevention and management of oral cancer (Table 1) [61]. Collating insights gleaned from prior investigations, it becomes increasingly apparent that oxidative stress and compromised antioxidant mechanisms serve as pivotal factors in the pathogenesis of oral cancer. Following a similar trajectory, a case–control examination conducted in southeastern China, with 382 oral cancer patients and 382 matched controls, revealed a significant association between dietary fiber, vitamin C intake, and oral cancer risk. Individuals with higher intakes of dietary fiber and vitamin C had substantially lower odds of developing oral cancer, with a 53% and 40% reduction in risk, respectively (Table 1) [62]. Healthcare providers may consider the integration of biomarker testing into OSCC screening and monitoring protocols. With the incorporation of biomarker data, diagnostic accuracy can be improved, and personalized treatment strategies can be informed. Biomarker testing including the measurement of nitric oxide and vitamin C levels has the potential to enhance the screening and monitoring of oral squamous cell carcinoma (OSCC) [65,66]. These biomarkers can help in early detection, facilitating timely intervention and improving patient outcomes [65]. However, the ideal serum biomarker for OSCC is yet to be discovered, and caution is needed in interpreting the available results [67]. Despite these challenges, the integration of biomarker data into comprehensive patient evaluations can enhance diagnostic accuracy and inform personalized treatment strategies [68].

In addition, in a clinical trial, melatonin, known for its potent antioxidant and oncostatic properties, was investigated as a potential adjunct therapy for OSCC. The study involved 50 OSCC patients; the administration of melatonin alongside NC led to reductions in miR-210 and CD44 expression. In addition, in a clinical trial, melatonin, known for its potent antioxidant and oncostatic properties, is explored as a potential adjunct therapy for squamous cell carcinoma of the oral cavity (OSCC). The study involved 50 OSCC patients, melatonin administration alongside neoadjuvant chemotherapy (NC) reduced miR-210 and CD44 expression, markers associated with tumor aggressiveness and chemoresistance. Furthermore, the assessment of clinical response, conducted using RECIST 1.1 criteria, demonstrated a notable decrease in tumor residue percentage post-melatonin treatment. The study underscores the potential of melatonin as a complementary therapeutic agent in OSCC management, particularly in modulating molecular markers associated with tumor aggressiveness and chemoresistance [63]. A multidisciplinary approach to cancer care including nutrition support is crucial for improving patient outcomes [69,70,71]. This approach should incorporate evidence-based interventions such as melatonin adjunct therapy and dietary modifications [64]. Collaboration with oncologists and nutritionists is essential for tailoring treatment plans to individual patient needs [70]. Personalized nutritional programs can help address the nutritional and metabolic disturbances that cancer patients may experience [72].

Experimental investigations have also demonstrated the protective effects of antioxidants against oral cancer. Hattori et al. (2022) [64] found that Canadian blueberry honey exhibited protective effects against H_2_O_2_-induced cytotoxicity in human buccal mucosal cells. These results underscore the potential of Canadian raspberry honey to prevent cellular oxidative stress, possibly through increasing the expression of HO-1 mRNA, and to promote wound healing by enhancing cell migration [64]. In addition to their preventive and therapeutic roles, antioxidants are emerging as promising markers for oral cancer diagnostics. Oxidant and antioxidant levels, particularly nitric oxide and vitamin C, are emerging as promising markers for oral cancer diagnostics. Juneja et al. found that the mean nitric oxide levels were significantly higher in OSCC patients (28.31 ± 5.57 μmol/L) compared to the healthy controls (11.11 ± 0.99 μmol/L). Meanwhile, the serum vitamin C levels were significantly reduced in OSCC patients (29.38 ± 4.08 μmol/L) compared to the healthy controls (45.76 ± 2.58 μmol/L) [73]. These findings suggest their potential utility as biomarkers in monitoring disease progression from precancerous lesions to cancer. Complementing these findings, a systematic review and meta-analysis sought to pool disparate datasets to unveil the role of pro-oxidants in the prognosis of oral cancer. The study revealed higher levels of MDA in the plasma, serum, and saliva samples of OSCC patients compared to healthy controls. These results demonstrate the potential of MDA as a diagnostic biomarker for OSCC, highlighting the need for further research to clarify its clinical utility [74].

The management of oral cancer therapy is riddled with challenges, particularly concerning side effects and patients [75]. This underscores the necessity for a multidisciplinary approach involving medical oncologists, dentists, and other oral healthcare specialists [76]. Nanomedicine presents constructive directions for therapy, offering potential in drug delivery and gene therapy applications [77]. Targeted drug delivery systems including nanoparticles and biomimetic systems are being explored for their capacity to improve oral cancer treatment outcomes [78]. Effectively addressing these challenges requires a thorough understanding of the intricate interactions among treatment modalities, tumor biology, and patient-related factors. Side effects such as mucositis, dysphagia, and xerostomia are prevalent and impact patient quality of life [75]. Ensuring patient adherence to treatment regimens is equally crucial, influenced by factors such as treatment-related toxicity, financial constraints, and psychosocial stressors [76].

## 4. Genetic Aspects related to Oxidative Stress and OSCC Biomarkers

Carcinogenesis is a complex, multi-step process that begins with abnormal oncogenic signals in various signaling pathways [79]. The debut and progression of cancer are closely linked to the generation of oxidative stress within cells [80]. The observation that a significant proportion of oxidative stress genes show a negative correlation with survival in solid carcinomas highlights the concept that oxidative stress plays a pivotal role in the biology of cancer cells [81].

Elevated levels of ROS disturb cellular homeostasis and contribute to the impairment of normal cellular functions. These disturbances are closely associated with the onset and advancement of different types of cancer [80]. In a 2017 study, researchers aimed to explore the role of ROS metabolism in the progression of OSCC. The study centered on evaluating the expression patterns of 84 genes associated with antioxidant metabolism in OSCC, aiming to uncover and comprehend the biological mechanisms linked to genes exhibiting differential expression in OSCC. The investigation identified twenty-one genes with significant differential expression in OSCC compared to non-tumor tissues. Among these, four genes showed upregulation, while 17 genes displayed downregulation in OSCC. Notably, the study introduced fourteen genes (ALOX12, CSDE1, DHCR24, DUOX1, DUOX2, EPHX2, GRLX2, GPX3, GSR, GSTZ1, MGTS3, OXR1, OXSR1, and SOD1) that exhibited differential expressions in OSCC for the first time. Through bioinformatics analysis, these differentially expressed genes were linked to fifteen crucial biological processes associated with carcinogenesis including inflammation, angiogenesis, apoptosis, genomic instability, invasion, survival, and cell proliferation. Moreover, the study indicated a significant involvement of glutathione metabolism in oral cancer, as four of these genes participate in this metabolic pathway (GSR, MGST3, GPX3, and GSTZ1). This underscores the potential importance of glutathione metabolism in the context of oral cancer development [82].

A biomarker represents any identifiable signature capable of quantifiably reflecting normal biological processes, pathological tissue changes, or responses to therapeutic interventions. Saliva biomarkers for OSCC can be classified into proteomic, transcriptomic, genomic, epigenomic, metabolic, and microbiota-based categories. The presence of distinct biomarkers in local secretions is a common trait across various malignancies, offering significant potential for early detection. Liquid biopsy, encompassing blood, body fluids like urine, and local secretions such as bronchial washings, pleural effusion, and ascitic fluid, has been extensively investigated as a viable diagnostic medium for malignancies. Similarly, saliva contains potential biomarkers capable of reflecting subtle molecular changes in tissue, showing promise for the early detection of OSCC [83].

Mitochondrial dysfunction stands out as a notable characteristic observed in oral cancer cells. Alterations in mitochondrial DNA (mtDNA) content are apparent in advanced head and neck squamous cell carcinoma, irrespective of age and smoking habits, and can potentially be detected in saliva. Primary changes in the mitochondrial genome include mutations and variations in copy numbers. Mitochondrial mutations can occur in both coding (such as cytochrome c oxidase genes) and noncoding regions, sometimes limited to a subset of mitochondrial DNA (mtDNA) copies among the 10^3^–10^4^ copies in each cell, known as heteroplasmy. The displacement loop (D-loop) has emerged as a hotspot for mutations in the mitochondrial genome, displaying strong biomarker potential. D-loop mutations may potentially impact the respiratory chain and bioenergetics. Mutated mitochondrial DNA (mtDNA) is significantly more abundant (19–220 times) than mutated p53 DNA and is easily accessible in bodily fluids such as saliva. In tobacco-betel quid chewers with OSCC, mitochondrial DNA copy numbers are notably high. Due to the substantial change in copy number, they can be examined in body fluids, making them well-suited as potent salivary markers for OSCC. The assessment of salivary mtDNA could be valuable for the early detection of OSCC and for monitoring treatment progress. Saliva samples have been utilized in various studies to identify P53 mutations. The DNA methylation status of tumor suppressor genes such as P16, death-associated protein kinase (DAPK), and methylguanine-DNA methyltransferase (MGMT) has been found to correlate with smoking history. Oral rinse samples demonstrated a strong correlation in reflecting this methylation status. Exposure to cigarette smoke diminishes the protective antioxidant capacity of saliva, rendering it highly deleterious. In this manner, saliva facilitates the catalysis of oral carcinogenesis, promoting its progression [78].

In oral cancer and potentially malignant disorders (OPMDs), indicators of lipid damage such as malondialdehyde (MDA) and 8-hydroxy-2-deoxyguanosine (8-OHdG), along with protein carbonyls, demonstrate an elevation. In a prior investigation conducted in 2010, heightened oxidative and nitrosative stress were observed in individuals with oral precancer and oral cancer. This was substantiated by increased levels of TNO (total nitric oxide) and MDA (malondialdehyde), indicating elevated lipid peroxidation due to nitric oxide. Furthermore, the antioxidant defenses were compromised, as evidenced by a reduction in the total antioxidant capacity (TAC). The cumulative evidence from the study including elevated levels of TNO and MDA in both serum and tissue, along with the positive correlation between TNO and MDA, suggests that oxidative DNA damage—an essential phenomenon in carcinogenesis—occurs due to the interplay between reactive ROS and reactive nitrogen species (RNS), along with the diminished total antioxidant capacity. Concurrently, levels of antioxidant vitamins (A, C, E), retinol, carotenes, and total antioxidant capacity are decreased in these conditions [60,83].

The current literature proposes over 100 potential biomarkers for OSCC. Among these, some promising candidates include oxidative stress responsive protein 1 (OXSR1), identified as a potential biomarker associated with oxidative stress, the total antioxidant capacity (TAC), an indicator of the overall antioxidant capacity, serving as a potential biomarker for OSCC, and metalloproteinase-9 (MMP-9), considered as another potential biomarker for OSCC, with a focus on its role in metalloproteinase activity. These biomarkers hold promise for further research and exploration in the context of OSCC diagnosis and monitoring. OXSR1, a 58-kD protein encoded by the OXSR1 gene, is widely expressed in various tissues. Studies indicate that OXSR1 plays a crucial role in essential cellular processes such as apoptosis, migration, and autophagy. Recent experimental data suggest its involvement in malignant progression, although its prognostic value and function in cancer remain unclear. MMP-9, a protease closely associated with cancer pathogenesis and progression, is implicated in the degradation of the extracellular matrix. Salivary MMP-9 levels showed significant differences between the OSCC patient group and the corresponding control group in a study from 2020. The findings from the same study indicated a noteworthy and negative correlation between the MMP-9 levels and OXSR1 levels. Additionally, the results demonstrated a significant positive correlation between the total antioxidant capacity (TAC) and OXSR1 [84].

A notable isoprostane, 8-iso-prostaglandin F2α (8-iso-PGF2α), also known as 15 F2t isoprostane/8-isoprostane, has emerged as a major isoprostane linked with oxidant injury. It is formed exclusively when endogenous antioxidants are depleted. Isoprostanes are increasingly acknowledged as a gold standard marker for oxidative stress. Their advantages over other markers encompass chemical stability, specificity as products of peroxidation, and detectable presence in all normal tissues and biological fluids including exhaled breath condensate. Importantly, their levels significantly rise in the presence of oxidant injury. Isoprostanes, particularly 8-isoprostane, have been employed as disease markers in conditions such as tissue fibrosis, prostate, and lung cancer. This underscores the potential utility of 8-isoprostane as a valuable marker in oral cancer and oral submucous fibrosis (OSF) [85].

Several studies have identified and pre-validated potential biomarkers for oral cancer diagnosis and prognosis including mRNAs and proteins such as IL-8 and SAT [86], CDH11, SPARC, POSTN, TNC, and TGM3 [87] as well as a range of salivary biomarkers [88]. However, the validation of potential biomarkers remains limited, posing challenges in their clinical translation. Emphasizing this limitation serves to highlight the critical gap in the current research and underscores the importance of further validation studies. The need for strong clinical validation of potential biomarkers in oral cancer diagnosis and prognosis has been emphasized. Wilhelm-Benartzi et al. [89] and Rivera et al. [90] stressed the pivotal role of preclinical work and stringent validation before incorporating biomarkers into clinical trials. Rivera et al. [90] further highlighted the potential of 41 biomarkers in oral squamous cell carcinoma, but emphasized the need for further validation in well-designed clinical cohort-based studies. Reid et al. [91] acknowledged the role of biomarkers in cancer clinical trials, but also noted the increased cost and burden on the research subjects. These studies collectively underscore the critical gap in current research and the importance of rigorous validation studies in this field.

## 5. Melatonin: Unraveling Its Potential in Oral Cancer Therapy and Beyond

Melatonin, a natural indoleamine known as N-acetyl-5-methoxytryptamine, is primarily synthesized by the mammalian pineal gland and various tissues including lymphocytes, the Harderian gland, liver, and the gastrointestinal tract. Accumulating evidence suggests that melatonin possesses antioxidant and oncostatic properties. Reports indicate that melatonin exhibits anticancer effects in various cancer types including breast cancer, lung cancer, colorectal cancer, gastric cancer, and cervical cancer. However, the precise mechanisms underlying melatonin’s anticancer effects on these cancers require further investigation [92]. Recent findings suggest that tumor cells play a role in inducing epigenetic changes in local neutrophils, influencing tumor progression. Neutrophils, traditionally overlooked in tumor biology, have been identified as potential modulators of tumor prognosis. Tumor-associated neutrophils (TANs) may exhibit either pro- or antitumoral effects. TANs have been linked to tumor regression by inducing tumor cell death through ROS production, expression of the apoptotic ligand TRAIL, and their ability to mediate antibody-dependent cell cytotoxicity. Moreover, TANs show increased transcription of cytokine and chemokine mRNAs compared to naive neutrophils. However, TANs have also been associated with promoting metastasis in squamous cell carcinoma. In oral squamous cell carcinoma, intense tumor-associated neutrophil (TAN) infiltration is positively correlated with advanced stage, lymphatic metastasis, and poor prognosis. In contrast, melatonin has been shown to reduce the survival and migration of TANs associated with oral squamous cell carcinoma. Melatonin achieves this by suppressing the TAN release of inflammatory factors (such as C-X-C motif chemokine ligand 8, C-C motif chemokine ligand 2, CCL4, and matrix metalloproteinase-9) through the blockade of the p38 MAPK and Akt signaling pathways. This suggests that melatonin may be beneficial in the treatment of squamous cell carcinoma by mitigating the migration, inflammatory responses, apoptosis resistance, pro-angiogenesis, and pro-motility effects of TANs. However, a detailed investigation is needed to unravel the intricate mechanisms of melatonin in the context of TANs in squamous cell carcinoma [93].

A study from 2020 focused on the effects of melatonin revealed the anticancer effects of melatonin in human tongue squamous cell carcinoma cells. Furthermore, melatonin appears as a promising candidate for anticancer therapy in tongue squamous cell carcinoma. Future research efforts are essential to further investigate the molecular mechanisms that support the anticancer effects of melatonin, thereby identifying innovative therapeutic targets for the treatment of human tongue squamous cell carcinoma [94].

Numerous recent cancer research studies have centered on melatonin (N-acetyl-5-methoxytryptamine) because of its acknowledged antioxidant, oncostatic, and anti-inflammatory properties. Melatonin’s involvement in regulating circadian rhythms has been extensively documented, and disrupted circadian patterns, often linked with factors like sleep deprivation, have been associated with various cancers affecting the head and neck, breast, prostate, and other systems. This suggests a potential connection between melatonin, circadian rhythms, and the development or progression of specific cancer types. Salivary melatonin (MLT) levels were discovered to be notably higher in patients with oral squamous cell carcinoma (OSCC) compared to those in healthy subjects. One conceivable explanation for the elevated levels of MLT in OSCC patients could stem from a potential disorder in MLT receptors within the affected tissue. This dysfunction might render OSCC cells insensitive to melatonin. This hypothesis suggests a direct protective effect of melatonin on healthy oral mucosa, potentially involving melatonin overexpression, insensitivity, or the reduced expression of MTNR1A (melatonin receptor 1A). Moreover, there could be the possibility of an undiscovered signaling pathway. Additionally, there might be an undiscovered signaling pathway involved. However, further research is essential to investigate the expression of melatonin receptors in vivo within OSCC tissue and clinically unchanged oral mucosa tissue in affected individuals. This research is crucial to either substantiate or refute these hypotheses [95].

Researchers are endeavoring to decipher the mechanisms by which melatonin influences cancer-related processes including its effects on oxidative stress, inflammation, and the regulation of cell growth. Additionally, delving into the interplay between melatonin and sleep patterns in the context of cancer may offer valuable insights into preventive and therapeutic strategies. Considering melatonin’s antioxidant and oncostatic properties, it has emerged as a promising focus for further exploration in cancer studies. Such investigations may contribute to the development of innovative treatment approaches or supportive therapies in the realm of cancer research [95].

## 6. Conclusions

The evidence overwhelmingly supports the involvement of oxidative stress in the development and progression of oral cancers. ROS generated through various pathways induce damage to cellular components, triggering carcinogenesis.

Understanding the intricate mechanisms by which oxidative stress contributes to oral cancer pathogenesis is crucial. From DNA damage and inflammation to alterations in cell signaling pathways, oxidative stress orchestrates a cascade of events that lead to malignant transformation.

Targeting oxidative stress pathways presents promising avenues for oral cancer management. Antioxidant therapies and lifestyle interventions aimed at reducing ROS levels could potentially mitigate cancer risk and improve treatment outcomes.

Oxidative stress interacts with other risk factors such as tobacco and alcohol consumption, exacerbating the carcinogenic process. Comprehensive approaches addressing these multifaceted interactions are warranted for effective prevention and treatment strategies.

Biomarkers associated with oxidative stress could serve as valuable diagnostic tools and prognostic indicators for oral cancers. Their incorporation into clinical practice may enhance early detection and facilitate personalized treatment strategies.

Continued research efforts are imperative to fully unravel the complexities of oxidative stress in oral cancers. Integrating cutting-edge technologies such as genomics and proteomics with translational studies holds the potential to revolutionize our understanding and management of this disease.

Our understanding of oxidative stress in oral cancer is well-elucidated. Specifically, we know that certain substances present in cells, modified by oxidation, can serve as markers [96]. However, the future research directions remain somewhat unclear. By defining explicit directions for future studies including the integration of advanced technologies such as genomics and proteomics, researchers can effectively fill the remaining gaps in knowledge and move the field forward. Several research teams have made recommendations for future studies. The adoption of new technologies, as proposed by Park and Owens (2006) [97] such as nanotechnology and targeted therapies, could fundamentally transform the approach to oral cancer diagnosis and treatment. Sarode et al. [98] suggested that advanced genomic technologies including transcriptomics, whole genome sequencing, and epigenomics, could shed light on the complex mechanisms behind cancer development via oxidative stress in the oral cavity. Through in-depth genomic analyses, researchers can uncover genetic changes, disrupted signaling pathways, and epigenetic alterations associated with oxidative stress-induced oral carcinogenesis. From another angle, Sachdev et al. [99] highlighted the importance of exploring ways to support antioxidant systems in people with oral cancer. By considering these directions for future research and taking advantage of technologies such as genomics and proteomics, scientists could answer more questions and significantly advance the field.

## Figures and Tables

**Table 1 biomedicines-12-01150-t001:** Antioxidant and prooxidant for diagnosis and treatment: insights from clinical trials, observational studies, and experimental research.

Study Design	Intervention	Outcome Measures	Main Findings	Conclusion	References
Observational Study	Measurement of total nitric oxide (TNO), malondialdehyde (MDA), and total antioxidant capacity (TAC) levels.	Levels of TNO, MDA, and TAC in serum and tissues; correlation between TNO and MDA levels.	Elevated TNO and MDA levels with concomitant depletion of TAC in serum and tissues of OSCC patients compared to controls. Positive correlation between TNO and MDA levels.	Findings suggest a deranged antioxidant defense system and heightened oxidative stress in OSCC patients, potentially contributing to oral carcinogenesis.	[60]
Observational cohort study	Assessment of serum iron levels.	Levels of serum iron.	Significant reduction in serum iron levels observed in the cancer group compared to the precancer group.	Alterations in serum iron levels may serve as a potential indicator of disease progression from precancer to cancer stages in oral malignancies.	[61]
	Assessment of serum selenium levels.	Levels of serum selenium.	Marked decrease in serum selenium levels observed in cancer group compared to precancer group.	Serum selenium levels exhibit significant variation across precancer and cancer stages, suggesting a potential role in disease pathogenesis and progression.	[61]
Case–control study	Evaluation of dietary fiber and vitamin C intake.	Association between dietary fiber, vitamin C intake, and oral cancer risk.	Intake of dietary fiber and vitamin C was significantly lower in oral cancer patients compared to control participants. Dietary intake of fiber and vitamin C were inversely related to the risk of oral cancer.	Dietary intake of fiber and vitamin C were inversely related to the risk of oral cancer.	[62]
Double-blinded RCT	Combination of melatonin and neoadjuvant chemotherapy (NC) (taxane, cisplatin, and 5-fluorouracil).	Clinical response evaluated using RECIST 1.1 criteria; measurement of miR-210 and CD44 expression before and after intervention.	Reduction in miR-210 and CD44 expression observed in both groups, but not statistically significant. Decrease in tumor residue percentage following melatonin treatment, although not statistically significant.		[63]
Experimental	Evaluation of the protective effects of different types of honey against H_2_O_2_^−^-induced cytotoxicity and enhancement of cell migration.	Assessment of H_2_O_2_^−^-induced cytotoxicity, mucosal cell migration, and expression of heme oxygenase-1 (HO-1) mRNA.	Canadian blueberry honey demonstrated a protective effect against H_2_O_2_^−^-induced cytotoxicity and enhanced cell migration. It increased the expression of HO-1 mRNA, indicating activation of antioxidant and cytoprotective mechanisms. The water-soluble components (>10 kDa) of blueberry honey showed cytoprotective effects, although their specific identity remains unknown.	Canadian blueberry honey shows promise in preventing H_2_O_2_^−^-induced oxidation and promoting wound healing, possibly through the activation of antioxidant enzymes like HO-1. These findings suggest the potential prophylactic and therapeutic benefits of blueberry honey in human stomatitis management.	[64]

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
