# Peer review of "Oxidative Stress in the Pathogenesis of Oral Cancer"

_biomedicines, 2024, doi:10.3390/biomedicines12061150_

Round 1
Reviewer 1 Report
Comments and Suggestions for Authors
The authors of the review touched upon a very important topic – the role of ROS metabolism in carcinogenesis. Though a lot of studies has been devoted to this topic, antioxidant therapy remains problematic. This is due to the very complex and heterogeneous participation of antioxidants in carcinogenesis. We study ROS quite a lot and know their role, but we study little about antioxidants, and yet the antioxidant network is no less complex and heterogeneous. The review contains very interesting and useful information and provides practical recommendations (especially Table 1). The data on melatonin as an oncostatic agent are very interesting. The review will be interesting from a scientific and practical point of view. However, while reading, I had a few questions. Below are comments that I think will improve the article. I hope the authors will take them into account.
Major comments
1. The title of the article seems too complicate to me. It can be simplified, for example, to “Oxidative stress in the pathogenesis of oral cancer”
2. Figure 1 seems to me too primitive, since it reflects only the simplified metabolism of the superoxide anion radical. The superoxide anion radical is not as important in carcinogenesis as hydrogen peroxide. It is hydrogen peroxide, as a secondary signaling messenger and a regulator of the redox state, that makes a significant contribution. The biochemistry of the glutathione system is also much broader and includes the participation of various redoxins. It seems to me that Figure 1 should either be removed or supplemented with metabolism of hydrogen peroxide and glutathione.
3. Section “Overview on Oxidative Stress, Cancer, and Antioxidants”. The authors very well described the role of ROS in carcinogenesis, pointing out the characteristics of the effect at each stage. The authors noted that the damaging or beneficial effects of ROS depend on the concentration. However, antioxidants also have different effects depending on the concentration, site of exposure and tumor stage. I would like the authors to highlight the following two points: 1) how an excess of antioxidants affects carcinogenesis, 2) of great importance is also the biochemical activity of the product of its transformation.
4. Table 1 should provide references to each study.
Minor comments
5. Please check subscript and superscript characters everywhere in chemical formulas (lines 53-56, table 1).
6. Please decrypt HIF (line 78).
7. The location of figure 1 is confused in relation to the legend.
8. The reference to Table 1 is missing in the text.
9. “Oxidative Stress-Related Biomarkers” in line 295 – is this the section heading?
10. The abbreviation OSCC appears several times (line 381, 383).
Author Response
Reviewer 1
Dear Reviewer,
Thank you for taking the time to review our paper entitled "Oxidative Stress in the Pathogenesis of Oral Cancer." We sincerely appreciate your positive feedback regarding the overall quality of our paper and are delighted that you found it interesting. Your recognition of our efforts is truly motivating and reinforces our commitment to excellence in research. We also want to express our gratitude for your constructive criticism regarding the biochemical by products of antioxidants and their effects on cancer. We have carefully considered your comments, have revised, and completed the manuscript accordingly. Our responses are given in a point-by-point manner below (in blue).
Respectfully yours
The authors of the review touched upon a very important topic – the role of ROS metabolism in carcinogenesis. Though a lot of studies has been devoted to this topic, antioxidant therapy remains problematic. This is due to the very complex and heterogeneous participation of antioxidants in carcinogenesis. We study ROS quite a lot and know their role, but we study little about antioxidants, and yet the antioxidant network is no less complex and heterogeneous. The review contains very interesting and useful information and provides practical recommendations (especially Table 1). The data on melatonin as an oncostatic agent are very interesting. The review will be interesting from a scientific and practical point of view. However, while reading, I had a few questions. Below are comments that I think will improve the article. I hope the authors will take them into account.
Major comments
- The title of the article seems too complicate to me. It can be simplified, for example, to “Oxidative stress in the pathogenesis of oral cancer”
Thank you for your comment. The title has been changed as requested
- Figure 1 seems to me too primitive, since it reflects only the simplified metabolism of the superoxide anion radical. The superoxide anion radical is not as important in carcinogenesis as hydrogen peroxide. It is hydrogen peroxide, as a secondary signaling messenger and a regulator of the redox state, that makes a significant contribution. The biochemistry of the glutathione system is also much broader and includes the participation of various redoxins. It seems to me that Figure 1 should either be removed or supplemented with metabolism of hydrogen peroxide and glutathione.
Thank you for your comment. The figure 1 has been removed
- Section “Overview on Oxidative Stress, Cancer, and Antioxidants”. The authors very well described the role of ROS in carcinogenesis, pointing out the characteristics of the effect at each stage. The authors noted that the damaging or beneficial effects of ROS depend on the concentration. However, antioxidants also have different effects depending on the concentration, site of exposure and tumor stage. I would like the authors to highlight the following two points: 1) how an excess of antioxidants affects carcinogenesis, 2) of great importance is also the biochemical activity of the product of its transformation.
Thank you for your comment. The 2 suggested points have been added to section “Overview on Oxidative Stress, Cancer, and Antioxidants”
The biochemical activity of antioxidant transformation products is equally im-portant. When antioxidants endure metabolic transformations in the body, they generate a diversity of metabolites with distinct biological activities. These by products result from reactions with radicals and hydroperoxides, and exhibit different antioxidant activities. Antioxidants, which include organic compounds of sulphur and phosphorus, undergo transformation during the hydroperoxide decomposition process. This process leads to the various oxidation products formation like sulphoxides, S-centred radicals and sul-phenic acids . Phenols and aromatic amines, on the other hand, trap alkyl and alkoxyl radicals by chain-breaking mechanisms. These antioxidants produce transformation products such as phenoxy and aminyl radicals, which then react to form alkylperoxycy-clohexadienones (R02-CHD) and other compounds [10,11]. The resulting transformation products shows anti cancer proprieties. Sulforaphane, a sulphoxide compound found in cruciferous vegetables, has shown potential in cancer prevention and treatment. It has been found to induce apoptosis, arrest cell growth, and modulate carcinogen metabolism [12,13]. Additionally, it has been shown to have anti-cancer effects, including protection from DNA damage and modulation of the cell cycle [14]. These effects are mediated, at least in part, by the activation of the transcription factor Nrf2 [15]. However, the influence of sulfonylurea derivatives, another type of sulfoxide, on cancer risk is less clear, with con-flicting results in studies [16]. In addition, A range of alkylperoxycyclohexadienones have been investigated for their potential in cancer therapy. Eibl et al. [17] and Singura et al.[18] both explored the antineoplastic properties of these compounds, with Eibl focusing on hexadecylphosphocholine and Sugiura on 5-alkylidene-4-hydroxy-2-cyclopentenones. Chen et al. [19] synthesized and evaluated a series of 2-alkylaminomethyl-5-(E)-alkylidene cyclopentanone hydrochlorides, finding them to be active against various human cancer cell lines. Veinberg et al. [20] further expanded the range of alkylperoxycyclohexadi-enones, synthesizing 6-alkylidenepenicillanate sulfones and related 3-alkylidene-2-azetidinones, which showed potent cytotoxic properties towards tumor cells.
Today, there is enough scientific evidence to point to the paradox of antioxidants. They have been considered a double-edged sword, capable not only to inhibit but also to promote carcinogenesis depending on their concentration, balance with pro-oxidants and context of use. While studies highlight the ability of antioxidants to restore oxidative bal-ance and protect against oxidative stress-induced DNA damage, other studies investigate the ability of antioxidants to fuel carcinogenesis. Banerjee et al. noted that the presence of N-acetyl cysteine (NAC), an exogenous antioxidant derived from the amino acid cysteine and commonly utilized as precursor in synthesizing glutathione, diminished the proapoptotic impact of andrographolide on colon cancer cell lines (T84 and COLO 205). The proapoptotic effect of andrographolide hinges on inducing endoplasmic reticulum (ER) stress through the production of reactive oxygen species (ROS) [21]. In addition, an antioxidants excess have been found to compromise the cancer cells elimination during radiation therapy by potentially inhibiting the mechanisms of cytotoxicity triggered by oxidative stress, which may influence the outcome of carcinogenesis [22]. Study by Sa-kamoto et al. indicate that vitamin E decreases the radiation effectiveness on both normal and malignant cells [23]. Similary, Witenberg et al. found that vitamin C alleviates radia-tion-induced apoptosis in cancer cells [24]. These findings suggest that antioxidants may hinder the intended cytotoxic effects of radiation on cancer cells. Furthermore, a study by Sayin et al. found that adding antioxidants to the diet accelerates lung cancer progres-sion by reducing ROS levels, preventing p53 activation and promoting tumour cell prolif-eration. In B-RAF- and K-RAS-induced lung cancer mouse models, supplementation with two antioxidants, N-acetylcysteine (NAC) and vitamin E, significantly increased tumour burden and reduced survival rates. RNA sequencing analysis revealed that despite their structural differences, both NAC and vitamin E induce closely coordinated alterations in tumour transcriptome profiles. These alterations are mainly characterised by an increase in the proliferation of cells expressing the wild-type form of the p53 protein, but not those with mutations in this protein [25].
- Table 1 should provide references to each study.
Thank you for your comment. The references have been added to the table
Minor comments
- Please check subscript and superscript characters everywhere in chemical formulas (lines 53-56, table 1).
Thank you for your comments. All the subscript and superscript characters have been checked
- Please decrypt HIF (line 78).
Thank you for your comment. HIF stands for Hypoxia-Inducible Factor
- The location of figure 1 is confused in relation to the legend.
Thank you for your comment. The figure 1 has been removed
- The reference to Table 1 is missing in the text.
Thank you for your comment. The references have been added to table 1
- “Oxidative Stress-Related Biomarkers” in line 295 – is this the section heading?
Thank you for your comment. It was an omission, we have removed the heading
- The abbreviation OSCC appears several times (line 381, 383).
Thank you for your comment. The repetition has been removed and the text has been modified
One conceivable explanation for the elevated levels of MLT in OSCC patients could stem from a potential disorder in MLT receptors within the affected tissue. This dysfunction might render OSCC cells insensitive to melatonin. This hypothesis suggests a protective direct effect of melatonin on healthy oral mucosa, potentially involving melatonin overexpression, insensitivity, or reduced expression of MTNR1A (melatonin receptor 1A). Moreover, there could be the possibility of an undiscovered signaling pathway. Additionally, there might be an undiscovered signaling pathway involved. However, further research is essential to investigate the expression of melatonin receptors in vivo within OSCC tissue and clinically unchanged oral mucosa tissue in affected individuals. This research is crucial to either substantiate or refute these hypotheses
Reviewer 2 Report
Comments and Suggestions for Authors
Dear authors, I read with great interest your manuscript titled “The Current State of Knowledge Regarding the Interactions Between the Oxidative Stress Status and Oral Cancer”.
The manuscript provides an overview of the relationship between oxidative stress status and oral cancer, focusing on mechanisms, biomarkers, therapeutic interventions, and clinical implications. The review integrates findings from epidemiological, clinical, and experimental studies to elucidate the complex interplay between reactive oxygen species (ROS), antioxidants, and oral cancer development and progression.
The manuscript thoroughly covers a wide aspect of oxidative stress and oral cancer, integrating evidence from epidemiological studies, clinical trials, and experimental research to support key points and arguments. The manuscript appropriately identifies knowledge gaps and areas for future research, highlighting the need for further investigation into the underlying mechanisms.
There are specific points that require a more thorough revision.
Critical Appraisal of the cited literature, including strengths, limitations, and potential biases of the studies cited. Additional details on the methodology of clinical trials and experimental studies referenced could be provided. The table needs significant refinement to specify the studies included and the level of evidence.
While the manuscript provides a comprehensive overview of genetic aspects and biomarkers related to oxidative stress in OSCC, it falls short in elucidating the underlying mechanistic pathways in sufficient detail. A more thorough exploration of the molecular mechanisms linking oxidative stress to oral cancer pathogenesis would enhance the scientific rigor and depth of the discussion.
The discussion on the clinical implications of the findings presented should be expanded, emphasizing their relevance to patient care, treatment decision-making, and outcomes.
The clinical validation of potential biomarkers is limited, and this should be more emphasized in the manuscript. The manuscript briefly discusses potential therapeutic interventions targeting oxidative stress pathways but lacks a detailed exploration of therapeutic strategies and their clinical implications. A more in-depth discussion on antioxidant therapies, lifestyle interventions, and emerging treatment modalities would provide valuable insights for clinicians and researchers.
While the manuscript outlines the current state of knowledge on oxidative stress in oral cancer, it could benefit from a more explicit delineation of future research directions. Providing clear recommendations for future studies, such as integrating advanced technologies like genomics and proteomics, would guide researchers in addressing remaining knowledge gaps and advancing the field.
Author Response
Reviewer 2
Dear Reviewer,
Thank you very much for all your notes, time, efforts, and support in improving our paper. Your recognition of our efforts to support key points and arguments with robust evidence is truly encouraging. We have carefully read the comments and have revised/ completed the manuscript accordingly. Our responses are given in a point-by-point manner below (in blue). To improve the quality of the manuscript, the text was modified, completed, corrected, and restructured.
Respectfully Yours
Dear authors, I read with great interest your manuscript titled “The Current State of Knowledge Regarding the Interactions Between the Oxidative Stress Status and Oral Cancer”. The manuscript provides an overview of the relationship between oxidative stress status and oral cancer, focusing on mechanisms, biomarkers, therapeutic interventions, and clinical implications. The review integrates findings from epidemiological, clinical, and experimental studies to elucidate the complex interplay between reactive oxygen species (ROS), antioxidants, and oral cancer development and progression. The manuscript thoroughly covers a wide aspect of oxidative stress and oral cancer, integrating evidence from epidemiological studies, clinical trials, and experimental research to support key points and arguments. The manuscript appropriately identifies knowledge gaps and areas for future research, highlighting the need for further investigation into the underlying mechanisms. There are specific points that require a more thorough revision. Critical Appraisal of the cited literature, including strengths, limitations, and potential biases of the studies cited. Additional details on the methodology of clinical trials and experimental studies referenced could be provided.
- The table needs significant refinement to specify the studies included and the level of evidence.
Thank you for you comment. The table has been modified as suggested
- While the manuscript provides a comprehensive overview of genetic aspects and biomarkers related to oxidative stress in OSCC, it falls short in elucidating the underlying mechanistic pathways in sufficient detail. A more thorough exploration of the molecular mechanisms linking oxidative stress to oral cancer pathogenesis would enhance the scientific rigor and depth of the discussion.
Thank you for your valuable comment. The molecular mechanisms linking oxidative stress to oral cancer pathogenesis has been added to the text (section 3 : Oxidative Stress and Antioxidant Mechanism in Oral Cancer Research: Insights from Epidemiological, Clinical and Experimental Studies)
Many studies have investigated the link between oxidative stress and oral cancer. This process is mediated through various mechanistic pathways that contribute to cancer initiation and progression. Over expression of enzymes like NOX, COX, LOX, and NOS, are significant sources of ROS/RNS in oral cancer [28-32] These species act as redox mes-sengers, impacting the activity of kinases/phosphatases and signal transduction path-ways , which in turn regulate transcription factors involved in cancer development. In ad-dition, ROS/RNS modulate the activity of a range transcription factors, including APE1/Ref-1, HIF-1α, AP-1, Nrf2, NF-κB, p53, FOXO, STAT, and β-catenin [32]. These fac-tors are essential in genes associated with cell proliferation regulation, survival, and apoptosis, thereby influencing cancer progression [32]. Furthermore, the canonical WNT/β-catenin pathway plays a significant role in oral cancer, influencing prognosis and treatment response [33, 34]. Its aberrant activation can lead to apoptosis resistance and tumorigenesis [34] while its role in oral carcinogenesis is well-documented [35]. The Wingless-Type-1 (WNT-1) pathway, a key component of this signaling, has been impli-cated in the progression and metastasis of oral cancer [36] Activation of WNT signaling begins with WNT ligand binding to Frizzled receptors and LRP5/6 co-receptors, triggering beta-catenin stabilization and its subsequent accumulation in the cytoplasm. Transloca-tion of beta-catenin into the nucleus forms a complex with TCF/LEF transcription factors, activating target genes like C-Myc, cyclin D1, c-jun, fra-1, and u-PAR, implicated in tumor progression. Dysregulation of this pathway in oral cancer correlates with heightened ag-gressiveness, metastatic propensity, and poorer prognosis, underscoring its significance as a therapeutic target for OSCC management [36,37,38].
- The discussion on the clinical implications of the findings presented should be expanded, emphasizing their relevance to patient care, treatment decision-making, and outcomes.
Thank you for your comment. The clinical implications of the findings presented have been added in the section « Oxidative Stress and Antioxidant Mechanism in Oral Cancer Research: Insights from Epidemiological, Clinical and Experimental Studies » after each relevant study as suggested
Diagnostic Biomarkers:
Healthcare providers may consider the integration of biomarker testing into OSCC screening and monitoring protocols. The incorporation of biomarker data, diagnostic ac-curacy can be improved, and personalized treatment strategies can be informed. Bi-omarker testing, including the measurement of nitric oxide and vitamin C levels, has the potential to enhance the screening and monitoring of oral squamous cell carcinoma (OSCC) [60,61]. These biomarkers can help in early detection, facilitating timely interven-tion and improve patient outcomes [60]. However, the ideal serum biomarker for OSCC is yet to be discovered, and caution is needed in interpreting available results [62] .Despite these challenges, the integration of biomarker data into comprehensive patient evalua-tions can enhance diagnostic accuracy and inform personalized treatment strategies [63]..
Clinical Studies and Trials (For melatonin)
A multidisciplinary approach to cancer care, including nutrition support, is crucial for improving patient outcomes [64,65,66]. This approach should incorporate evidence-based interventions such as melatonin adjunct therapy and dietary modifications [64]. Collabo-ration with oncologists and nutritionists is essential for tailoring treatment plans to indi-vidual patient needs [65]. Personalized nutritional programs can help address the nutri-tional and metabolic disturbances that cancer patients may experience [67].
Oxidative Stress and Antioxidant Defenses:
The understanding of the oxidative stress role in oral carcinogenesis highlights the anti-oxidant-based interventions potential in preventing, inhibiting, and even reversing the OSCC progression. The potential for antioxidant-based interventions in preventing, inhib-iting, and even reversing OSCC progression is highlighted, with a focus on the role of an-tioxidant phytochemicals in periodontal disease [52]. Regular monitoring of antioxidant levels and oxidative stress markers is recommended to assess disease progression and treatment response [6]. Healthcare providers may consider incorporating antioxidant supplementation or dietary recommendations into OSCC management strategies [53].
Risk Factors and Mechanisms
Healthcare providers are key figures in educating patients about the dangers linked to to-bacco and alcohol consumption concerning OSCC. Promoting smoking cessation and moderating alcohol intake can substantially decrease the risk of OSCC. Additionally, reg-ular oral health screenings should be emphasized for early detection and intervention. Some studies emphasize the importance of this education, specifically highlighting the personalized reports effectiveness in reducing alcohol-related risks [42,43]. Patton et al. [44] underscores the need for training in tobacco and alcohol cessation, particularly among dental providers. Shaik et al. [45] further underscore the oral health professionals importance in advocating for tobacco-free lifestyles and implementing intervention pro-grams.
The clinical validation of potential biomarkers is limited, and this should be more emphasized in the manuscript.
Thank for your comment. The clinical validation of potential biomarkers has been added to the end of section « Genetic Aspects related to Oxidative Stress and OSCC Biomarkers »
Several studies have identified and pre-validated potential biomarkers for oral cancer diagnosis and prognosis, including mRNAs and proteins such as IL-8 and SAT [81], CDH11, SPARC, POSTN, TNC, and TGM3 [82], and a range of salivary biomarkers [83]. However, validation of potential biomarkers remains limited, posing challenges in their clinical translation. Emphasizing this limitation serves to highlight the critical gap in current research and underscores the importance of further validation studies. The need for strong clinical validation of potential biomarkers in oral cancer diagnosis and progno-sis has been emphasized. Wilhelm-Benartzi et al. [84] and Rivera et al. [85] stress the piv-otal role of preclinical work and stringent validation before incorporating biomarkers into clinical trials. Rivera et al. [85] further highlights the 41 biomarkers potential in oral squamous cell carcinoma, but highlights the need for further validation in well-designed clinical cohort-based studies. Reid et al. [86] acknowledges the role of biomarkers in can-cer clinical trials, but also notes the increased cost and burden on research subjects. These studies collectively underscore the critical gap in current research and the importance of rigorous validation studies in this field.
- The manuscript briefly discusses potential therapeutic interventions targeting oxidative stress pathways but lacks a detailed exploration of therapeutic strategies and their clinical implications.
Thank you for your comment. The exploration of therapeutic strategies and their clinical implications have been added to « Oxidative Stress and Antioxidant Mechanism in Oral Cancer Research: Insights from Epidemiological, Clinical and Experimental Studies section » as sugggested.
The management of oral cancer therapy is riddled with challenges, particularly con-cerning side effects and patient [70]. This underscores the necessity for a multidisciplinary approach involving medical oncologists, dentists, and other oral healthcare specialists [71]. Nanomedicine presents a constructive directions for therapy, offering potential in drug delivery and gene therapy applications [72]. Targeted drug delivery systems, in-cluding nanoparticles and biomimetic systems, are being explored for their capacity to improve oral cancer treatment outcomes [73]. Effectively addressing these challenges re-quires a thorough understanding of the intricate interactions among treatment modalities, tumor biology, and patient-related factors. Side effects such as mucositis, dysphagia, and xerostomia are prevalent and impact patient quality of life [70]. Ensuring patient adher-ence to treatment regimens is equally crucial, influenced by factors such as treat-ment-related toxicity, financial constraints, and psychosocial stressors [71].
- A more in-depth discussion on antioxidant therapies, lifestyle interventions, and emerging treatment modalities would provide valuable insights for clinicians and researchers.
Thank you for your comment. A more in-depth discussion on antioxidant therapies, lifestyle interventions, and emerging treatment modalities has been added to section « Oxidative Stress and Antioxidant Mechanism in Oral Cancer Research: Insights from Epidemiological, Clinical and Experimental Studies »
Lifestyle interventions significantly impact cancer treatment outcomes, both posi-tively and negatively [39].
Smoking is strongly associated with various cancers development, with 9 out of10 lung cancer deaths. Studies indicates that smoking can alter the anti-cancer medications metabolism, potentially leading to higher clearance rates of specific drugs like erlotinib, which could diminish treatment efficacy. In addition, smoking may also affect the iri-notecan metabolism, an antineoplastic agent that acts as a specific inhibitor of DNA topoisomerase, leading to lower concentrations of its active metabolite, SN-38, and poten-tially altering treatment outcomes [39]. Another treatment disturbing factor is the alcohol. Moderate alcohol consumption has benn link toreduced overall mortality, but heavy al-cohol intake is associated with a higher risk of developing specific cancers, including head and neck cancer and esophageal carcinoma. Similar to smoking, continued alcohol use after a cancer diagnosis or treatment is associated with a higher risk of disease recur-rence or second malignancy. Alcohol consumption can influence the activity of drug-metabolizing enzymes, particularly CYP3A, a a human gene locus responsible for the metabolism of more than 50% of all therapeutic medications that undergo phase I me-tabolism, which may impact the metabolism and efficacy of certain anti-cancer drugs [39]. Healthcare providers are key figures in educating patients about the dangers linked to to-bacco and alcohol consumption concerning OSCC. Promoting smoking cessation and moderating alcohol intake can substantially decrease the risk of OSCC. Additionally, reg-ular oral health screenings should be emphasized for early detection and intervention. Some studies emphasize the importance of this education, specifically highlighting the personalized reports effectiveness in reducing alcohol-related risks [42,43]. Patton et al. [44] underscores the need for training in tobacco and alcohol cessation, particularly among dental providers. Shaik et al. [45] further underscore the oral health professionals importance in advocating for tobacco-free lifestyles and implementing intervention pro-grams.
In the other side, lifestyle interventions that include exercise and dietary advice have shown feasibility and positive impacts on health outcomes such as exercise behavior, fa-tigue, and functional capacity in cancer survivors [46]. In their single arm feasibility study, Karianne et al. aimed to evaluate the feasibility of a 12-month lifestyle intervention pro-gram (like diet, physical activity, stress management, and smoking cessation), for cancer patients undergoing chemotherapy. Participants were newly diagnosed cancer patients receiving chemotherapy with curative or palliative intent at a cancer treatment center in Norway. Out of 161 eligible patients, 100 agreed to participate, with a completion rate of 38%. Dropouts were associated with older age, certain diagnoses, and smoking status. A significant dietary habits, distress levels, and physical activity improvements was rec-orded by the finding as well as the feasibility and potential benefits of a comprehensive lifestyle intervention for cancer patients undergoing chemotherapy, including those in palliative care [47]. In addition, Godsey et al. explores the potential synergy between herb-al supplements and conventional treatments. For instances, anthocyanins extracted from Black Raspberries have exhibited notable cancer-preventing properties in both laboratory and animal studies. Application of an anthocyanin gel to the oral cavity has shown promise in suppressing genes associated with cancer cell proliferation, as well as inhibit-ing esophageal tumorigenesis and providing protection against benzo(a)pyrene-induced transformation [48]. Green tea polyphenols have been also extensively studied for their chemopreventive potential. They trigger apoptosis, induce cell cycle arrest, and block the epidermal growth factor receptor tyrosine kinase phosphorylation. (-)-Epigallocatechin-3-gallate (EGCG), a green tea polyphenols, have shown promising results in inhibiting carcinogenesis in animal models [49]. These polyphenols have also been linked to a reduced risk of various cancers, including oral cancer, in epidemiological studies [50]. In a mouse model of prostate cancer, oral consumption of green tea polyphe-nols was found to inhibit insulin-like growth factor-I-induced signaling, suggesting a po-tential mechanism for their anti-cancer effects [51]. These studies collectively suggest that lifestyle interventions, including the avoidance of harmful habits and the use of comple-mentary treatments, can significantly impact the effectiveness of cancer treatment.
- While the manuscript outlines the current state of knowledge on oxidative stress in oral cancer, it could benefit from a more explicit delineation of future research directions. Providing clear recommendations for future studies, such as integrating advanced technologies like genomics and proteomics, would guide researchers in addressing remaining knowledge gaps and advancing the field.
Thank you for your comment. recommendations for future studies, such as integrating advanced technologies like genomics and proteomics have been added to section « conclusion »
Our understanding of oxidative stress in oral cancer is well elucidated. Specifically, we know that certain substances present in cells, modified by oxidation, can serve as markers [91]. However, future research directions remain somewhat unclear. By defining explicit directions for future studies, including the integration of advanced technologies such as genomics and proteomics, researchers can effectively fill the remaining gaps in knowledge and move the field forward. Several research teams have made recommenda-tions for future studies. The adoption of new technologies, as proposed by Park and Owens, 2006 [92], such as nanotechnology and targeted therapies, could fundamentally transform the approach to oral cancer diagnosis and treatment. Sarode et al. [93] suggests that advanced genomic technologies, including transcriptomics, whole genome sequen-cing, and epigenomics, could shed light on the complex mechanisms behind cancer development via oxidative stress in the oral cavity. Through in-depth genomic analyses, researchers can uncover genetic changes, disrupted signalling pathways and epigenetic alterations associated with oxidative stress-induced oral carcinogenesis. From another angle, Bhat et al. [94] highlights the importance of exploring ways to support antioxidant systems in people with oral cancer. By considering these directions for future research and taking advantage of technologies such as genomics and proteomics, scientists could answer more questions and significantly advance the field.
Kind regards,
Round 2
Reviewer 2 Report
Comments and Suggestions for Authors
Dear authors,
thank you for amending the manuscript. I do not have any further comments.
Author Response
Thank you.
Kind regards,